# Mixed signals: interpreting mixing patterns of different soil bioturbation processes through luminescence and numerical modelling

Willem Marijn van der Meij[1], Svenja Riedesel[1,2], Tony Reimann[1]

[1] Institute of Geography, University of Cologne, Zülpicher Straße 45, 50674 Cologne, Germany

[2] Department of Physics, Technical University of Denmark, Frederiksborgvej 399, 4000 Roskilde, Denmark

*Correspondence to*: W. Marijn van der Meij (m.vandermeij@uni-koeln.de)

**Abstract.** Soil bioturbation plays a key role in soil functions such as carbon and nutrient cycling. Despite its importance, fundamental knowledge on how different organisms and processes impact the rates and patterns of soil mixing during bioturbation is lacking. However, this information is essential for understanding the effects of bioturbation in present-day soil functions and on long-term soil evolution.

Luminescence, a light-sensitive mineral property, serves as a valuable tracer for long-term soil bioturbation over decadal to millennial timescales. The luminescence signal resets (bleaches) when a soil particle is exposed to daylight at the soil surface and accumulates when the particle is buried in the soil, acting as a proxy for subsurface residence times. In this study, we compiled three luminescence-based datasets of soil mixing by different biota and compared them to numerical simulations of bioturbation using the soil-landscape evolution model ChronoLorica. The goal was to understand how different mixing processes affect depth profiles of luminescence-based metrics, such as the modal age, width of the age distributions and the fraction of bleached particles.

We focus on two main bioturbation processes: mounding (advective transport of soil material to the surface) and subsurface mixing (diffusive subsurface transport). Each process has a distinct effect on the luminescence metrics, which we summarized in a conceptual diagram to help with qualitative interpretation of luminescence-based depth profiles. A first attempt to derive quantitative information from luminescence datasets through model calibration showed promising results, but also highlighted gaps in data that must be addressed before accurate, quantitative estimates of bioturbation rates and processes are possible. The new numerical formulations of bioturbation, which are provided in an accompanying modelling tool, provide new possibilities for calibration and more accurate simulation of the processes in soil function and soil evolution models.

Keywords: Bioturbation, luminescence, soil evolution, numerical modelling

# 1 Introduction

Bioturbation is the umbrella term for soil mixing processes by various organisms. Bioturbation plays a key role in soil nutrient cycling, carbon sequestration, erosion, and the redistribution of contaminants and pollutants (Wilkinson et al., 2009; Briones, 2014; Creamer et al., 2022). Despite its pivotal role in regulating soil functions, we have an incomplete understanding regarding how different organisms and ecosystems impact the types and rates of mixing processes, how these rates vary with soil depth and how different mixing processes interact within the soil (Schiffers et al., 2011; Michel et al., 2022). These insights are essential for accurately modelling the effects of bioturbation on present-day soil functions and the long-term evolution of soils (Creamer et al., 2022; Meng et al., 2022).

In this work, we examine two key soil bioturbation processes: mounding and subsurface mixing (Wilkinson et al., 2009). *Mounding* is the upward advective transport of soil material, which is deposited on the surface in mounds and later eroded and buried by newly mounded material. *Subsurface mixing* involves the diffusive up- and downward exchange of soil material throughout the entire soil profile at various depths. Many soil organisms display both processes in different capacities, depending on their feeding and burrowing behaviour. Gophers and mound-building termites such as *Macrotermes* are mainly known for mounding ( Gabet, 2000; Kristensen et al., 2015), while organisms that mainly reside in the subsurface, such as gallery-building ants, endogeic earthworms and tree roots, typically show subsurface mixing behaviour (Richards, 2009; Halfen and Hasiotis, 2010; Taylor et al., 2019). Anecic earthworms and *Aphaenogaster* ants, that visit the surface and create deep vertical burrows and galleries, contribute to both mounding and subsurface mixing (Bouché, 1977; Richards, 2009). Bioturbation is thus often an interplay of mounding and subsurface mixing, driven by various organisms, but also by environmental and climatic factors (Wilkinson et al., 2009; Kraus et al., 2022), which leads to mixed bioturbation signals in the soil. Although subsurface mixing is generally considered the dominant process, there is a lack of data or methods to differentiate the effects of both bioturbation processes (Wilkinson et al., 2009; Halfen and Hasiotis, 2010; Michel et al., 2022). Luminescence emitted by quartz and feldspar grains has successfully been used as tracer for bioturbation (Heimsath et al., 2002; Madsen et al., 2011; Stockmann et al., 2013; Johnson et al., 2014; Gliganic et al., 2015; Hanson et al., 2015; Reimann et al., 2017; Román-Sánchez et al., 2019a; Zhang et al., 2024). The luminescence signal accumulates over time due to ionizing radiation coming from naturally occurring radionuclides in the soil (uranium and thorium decay chains and potassium-40) and from cosmic rays. The luminescence signal is reset (bleached) when a soil particle is exposed to daylight. Thus, the luminescence signal is a proxy for the residence time of soil particles in the subsurface and is ideally measured on single grains when used as a tracer for soil mixing (Duller, 2008). The distribution of the luminescence signal of different grains in a sample informs about the type and intensity of the mixing process (Bateman et al., 2003, 2007). Moreover, their changes with depth provide additional information on rates, patterns and intensity of bioturbation.

Luminescence signals are often used in combination with numerical or analytical tools to calculate particle ages and soil mixing rates, and characterize mixing patterns (Furbish et al., 2018a, b; Román-Sánchez et al., 2019b; Schiffers et al., 2011; Gray et al., 2020; Yates et al., 2024). These tools are often based on a single diffusion-based implementation of the mixing process,

which limits the possibilities to separate mixing signals by different biota (Schiffers et al., 2011), or are based on models stemming from aquatic ecology without adequate testing for terrestrial environments (Michel et al., 2022). Recent developments in soil-landscape evolution modelling enable the integration of luminescence tracers with process-based simulations of soil and landscape processes (ChronoLorica model; van der Meij et al., 2023). This integration enables the simulation of the effects of different bioturbation processes on luminescence-depth profiles, which can help to quantify the impacts of different bioturbation processes on soil mixing, better formulate bioturbation processes and their effects on nutrient cycling and other soil functions (Creamer et al., 2022), simulate soil mixing over different spatial and temporal scales (e.g., Schiffers et al., 2011) and better represent the role of biota in soil-landscape evolution models (Meng et al., 2022).

The objective of this study is to provide qualitative and quantitative tools for differentiating the impacts of mounding and subsurface mixing during soil bioturbation using luminescence tracers. By integrating experimental luminescence-based bioturbation datasets with soil evolution modelling, we aim to 1) characterize typical luminescence-depth profiles for mounding and subsurface mixing, 2) determine how varying parameters and combinations of these processes affect these depth profiles and 3) derive quantitative process rates and contributions from experimental data through model calibration.

## 2    Methods

### 2.1    Conceptual models of soil mixing

Mounding and subsurface mixing have distinct effects on the soil and luminescence tracers. In this section, we conceptually discuss these effects as a basis for their numerical implementation.

Soil bioturbation by *mounding* causes a net upward transport of soil material to the soil surface (Figure 1a). This soil material is mined from previously buried material from the upper part of the soil (~ 1 m for termites, Kristensen et al., 2015), effectively leading to recycling of soil material in the mounding process over longer timescales. This recycling exposes a large part of the soil grains to daylight, leading to only a limited amount on non-surfaced grains that can carry a saturated luminescence signal. Typical mounding organisms are gophers, moles and termites (Gabet, 2000; Wilkinson et al., 2009; Kristensen et al., 2015). Mounding rates most likely decrease with depth due to decreasing biotic activity (Gray et al., 2020).

The diffusion-like transport caused by organisms that perform *subsurface mixing* moves soil material in between subsurface layers (Figure 1b). Typical mixing organisms are endogeic and anecic earthworms which (partly) live underground (Taylor et al., 2019), ants and subterranean termites that create subsurface galleries (Richards, 2009; Halfen and Hasiotis, 2010; Rink et al., 2013; Stewart and Anand, 2014; Taylor et al., 2019) and tree roots that shift material around when growing and which leave pores that can be filled with material after decay of the root material (Johnson et al., 2014; Ruiz et al., 2015). With subsurface mixing, there is much smaller proportion of grains that is transported to the surface, leaving a higher proportion of non-surfaced grains. Also, for subsurface mixing, mixing rates probably decrease with depth, due to decreased biotic activity (Figure 1b).

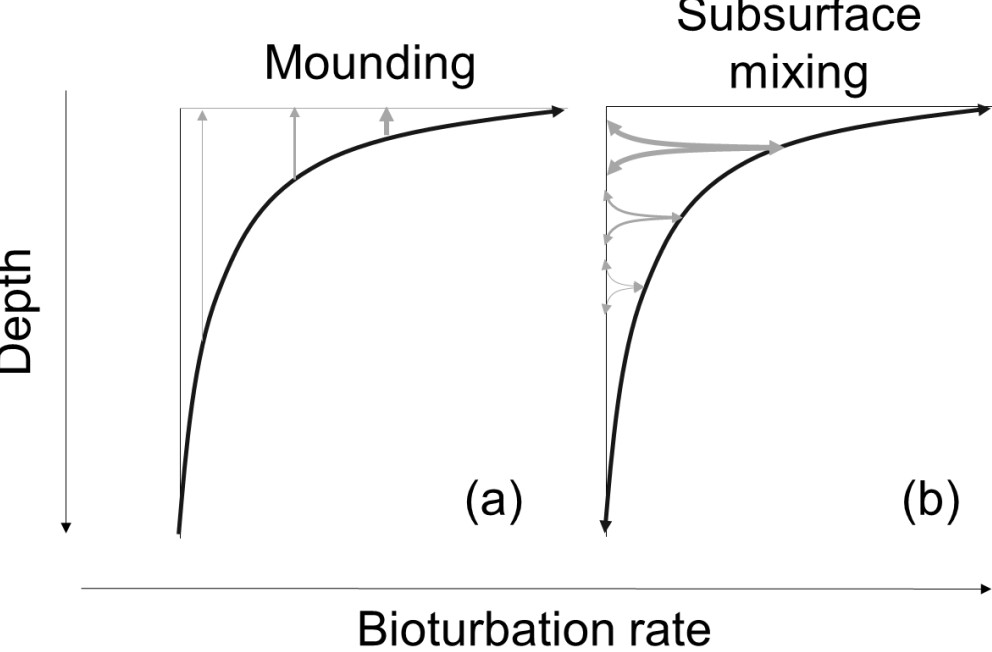

**Figure 1: Conceptual drawing of (a) mounding and (b) subsurface mixing. Subsurface mixing and mounding are visualized here with an exponential depth function (see Sect. 2.3.2). The arrows indicate direction and their thickness the intensity of soil transport.**

## 2.2 Experimental studies

We compiled three quartz and feldspar single-grain luminescence-based datasets of soil mixing by different organisms to characterize luminescence-depth profiles (Table 1). The main bioturbating organisms are termites who preferentially mound

(Kristensen et al., 2015), anecic earthworms who both mound and mix the subsurface (Von Suchodoletz et al., 2023) and ants who mainly mix the subsurface (Román-Sánchez et al., 2019a). All measurements were performed using Risø TL/OSL DA15 and DA20 luminescence readers equipped with $^{90}Sr/^{90}Y$ beta sources. The luminescence signals of single-grain quartz for the termites study were stimulated using a green laser for single grains (Kristensen et al., 2015). The signals were detected by a UV-sensitive photomultiplier tube (PMT) through a 7.5 mm Schott U-340 filter. K-rich feldspars were stimulated using an IR

laser and the signals were detected with a LOT/ORIEL D410 interference filter (Román-Sánchez et al., 2019a; Von Suchodoletz et al., 2023). Details regarding the sample preparation and the exact measurement conditions are given in the respective publications and a summary is provided in Table 1.

**Table 1: Overview of experimental single-grain luminescence datasets used in this study. When reported in the original publications, species names, climate zones and ecosystems are mentioned. Q = Quartz, FSP = Feldspar, SG = single grains, Post-IR IRSL = Post-infrared infrared stimulated luminescence, PH = preheat**

| Organism | Primary mixing process | Climate zone and ecosystem | Reference | Selected profile | Active mixing depth | Defined bioturbation period or saturation criteria | Luminescence method |
|---|---|---|---|---|---|---|---|
| Termites (*Macrotermes natalensis*) | Mounding | Tropical (Aw), Savannah | (Kristensen et al., 2015) | Unit II | 1.02 m | < 4 ka, onset of deforestation and start of savannah ecosystem | Q, SG, OSL, grain size: 90-180 µm, OSL |
| Anecic earthworms | Subsurface mixing and mounding | Warm-summer humid continental (Dfb), ecosystem not reported | (Von Suchodoletz et al., 2023) | Profile 2 | 60 cm | < 13.2 ka, estimated start of bioturbation by earthworms > 3.8 ka, end of bioturbation, due to burial of soil below burial mound | FSP SG, post-IR$_{50}$IRSL$_{150}$ (PH 175 °C, 60 s), grain size: 212-250 µm |
| Ants | Subsurface mixing | Mediterranean climate (BSk), Oak-woodland savannah | (Román-Sánchez et al., 2019a) | SC-10 | 50 cm | 2*D$_0$ (Wintle and Murray, 2006) | FSP SG, post-IR$_{50}$IRIRSL$_{175}$ (PH 200 °C, 60 s), grain size: 212-250 µm |

From these datasets, we are only interested in the ages of grains that have been bioturbated by the current dominant bioturbating agent. For the termites and worms datasets, there is a defined time period in which the current agent has been and continues to be active (Table 1). Grains falling outside of this timeframe are filtered out and excluded from our analysis of age distributions. Instead, we incorporate this fraction of particles ($f_{filtered}$) with the fraction of grains that have not reached the surface at all and have a saturated luminescence signal ($f_{non-surfaced}$). The remaining fraction ($f_{bio}$) contains the grains that have reached the surface through bioturbation by the current dominant agent (Eq. (1)). $f_{bio}$, or the bioturbated fraction, is similar to the non-saturation factor (NSF) as defined by Reimann et al. (2017), with the addition of another rejection criterion based on the bioturbation period.

$$f_{bio} = 1 - \left( f_{non-surfaced} + f_{filtered} \right) \tag{1}$$

## 2.3 Simulations

### 2.3.1 Model description

The bioturbation simulations are performed in the model ChronoLorica (Van der Meij and Temme, 2022; Van der Meij et al., 2023), which is an extension of the soil-landscape evolution model Lorica (Temme and Vanwalleghem, 2016; Van der Meij et al., 2020). Lorica is a mass-based four-dimensional numerical model that simulates the development of terrain and soil properties due to various geomorphic and pedogenic processes. The landscape is represented by a raster, where every raster

cell contains a pre-defined number of soil layers. The layers contain a mass of five mineral soil textures (coarse, sand, silt, clay and fine clay) and two organic matter types (young and old). Throughout the simulations, the contents of the layers change due to the addition, removal or transformation of the soil material by the simulated processes. At this stage, the model is

insensitive to parent material variations, as it does not include grain-size dependent mixing rates. Changes in the mass composition of each layer are translated to changes in layer thickness and surface elevation through the bulk density. Lorica works with dynamic layer thicknesses, enabling easy calculation of additions and subtractions from each layer. The layers start with a pre-defined initial thickness. When a layer thickness becomes more than 55% thicker than the initial thickness, the layer splits into two new layers. When a layer thickness becomes thinner than 55% of the initial thickness, the layer is merged with

a neighbouring layer. Due to its coarse spatial resolution and temporal resolution, the model is not suitable to simulate pore size dynamics due to bioturbation.

The ChronoLorica extension couples the pedogenic and geomorphic processes in the model to several geochronometers. In this study, we use soil particle burial ages, akin to luminescent grains, as tracer for bioturbation. We term these *luminescence particles* in this study. These particles do not have specific dimensions, but should be considered as objects that carry a specific

age that is analogous to a luminescence age. This age increases during their time of burial and resets when the particles are transported into the surface layer. This surface layer has a fixed depth that represents the bleaching depth. The transport of luminescence particles is coupled to the transport of the sand fraction of the model, because the sand fraction is the texture class that is typically used for single-grain luminescence dating (Duller, 2008). Due to memory constraints in the model, the number of tracked luminescence particle ages is much lower than the number of sand particles present in each layer. Therefore,

we used a probabilistic approach to determine whether a luminescence particle is transported together with the sand from one layer to another. The transport probability for each individual particle is determined by dividing the transported mass of sand out of a source layer by the total mass of sand present in that source layer (Eq. (2)).

$$P_{transport} = \frac{sand\ transported\ [kg]}{total\ sand\ present\ [kg]} \tag{2}$$

### 2.3.2 Depth functions

Bioturbation is most likely a depth-dependent process, but whether the mixing rates decrease linearly or exponential with depth is still unknown (Gray et al., 2020). In our simulations, we consider three typical depth functions that describe how mixing intensity changes with increasing soil depth (Minasny et al., 2016; Figure 2). These depth functions can be applied to both bioturbation processes. The depth functions describe i) a linear decrease with depth (*gradational*, $df_{grd}(z)$, Eq. (3)), ii) an exponential decrease with depth (*exponential*, $df_{exp}(z)$, Eq. (4)) and iii) a uniform mixing rate, which reduces abruptly to zero

below the mixing zone (*abrupt*, $df_{abr}(z)$, Eq. (5)). The depth decay parameters ($dd_{grd}$, $dd_{exp}$, $dd_{abr}$) [m$^{-1}$] determine the shape and gradient of the depth functions.

$$df_{grd}(z) = \begin{cases} -dd_{grd} * z, \ z \leq \dfrac{1}{dd_{grd}} \\ 0, \ z > \dfrac{1}{dd_{grd}} \end{cases} \qquad (3)$$

$$df_{exp}(z) = 1 - e^{-dd_{exp}*z} \qquad (4)$$

$$df_{abr}(z) = \begin{cases} 1, \ z \leq dd_{abr} \\ 0, \ z > dd_{abr} \end{cases} \qquad (5)$$

The total bioturbation occurring in the soil profile, $BT_{pot}$ [kg m$^{-2}$ a$^{-1}$], is distributed across all soil layers using one of the depth functions (Eqs. (3-5)) and depths of each layer (Eq. (6)). To determine how much bioturbation occurs in a particular layer, the depth function $df(z)$ is integrated between the upper and lower depths of that layer ($z_{upper}$, $z_{lower}$). This value is divided by the integral of the depth function $df(z)$ over the entire active mixing depth. The resulting fraction is multiplied by $BT_{pot}$ to calculate the bioturbation occurring in that specific layer ($BT_{layer}$, [kg m$^{-2}$ a$^{-1}$]). The limit of the active mixing depth is indicated with the

parameters $z_{lim}$, which is $1/dd_{grd}$ for the gradational function, the total soil depth $sd$ for the exponential function, where $z \leq sd$, and $dd_{abr}$ for the abrupt function.

$$BT_{layer} = BT_{pot} * \frac{\int_{z_{upper}}^{z_{lower}} df(z)dz}{\int_{0}^{z_{lim}} df(z)dz} \qquad (6)$$

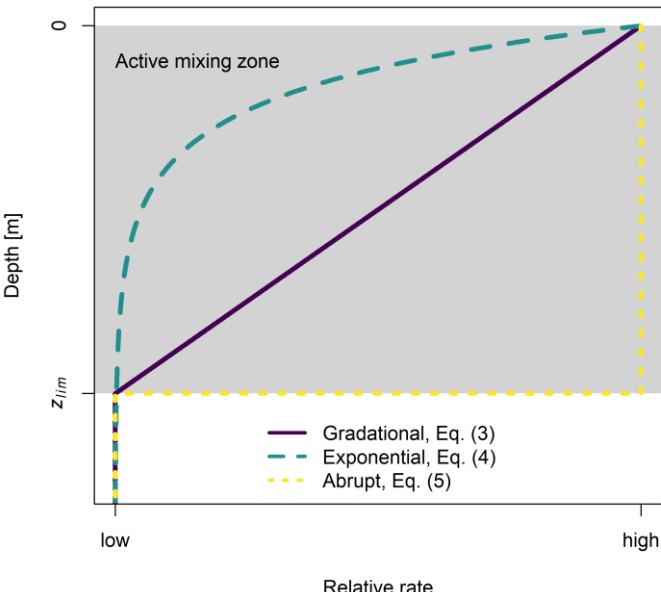

**Figure 2: Depth functions that are used for bioturbation simulations. The depth functions determine how bioturbation rates change**
**with soil depth. The parameters are selected, so that all bioturbation effectively occurs in the active mixing zone (grey area), ranging from the surface to the depth of $z_{lim}$.**

### 2.3.3    Process descriptions

We simulate the mounding process as upward transport of soil material from the subsurface. Eq. (6) determines how much material is taken from each soil layer. This material is then transported to the surface layer, gradually burying previously mounded material. In this implementation, the development and erosion of surface mounds is simplified into generation of a new surface layer, that results from the mound erosion.

The subsurface mixing process is simulated by an exchange of soil material between all present soil layers (Figure 1b). Eq. (6) determines how much material each soil layer (donor layer) can exchange in total with all other exchange layers in the profile. The exchange $BT_{exchange}$ between the donor layer and the exchange layers is controlled by Eq. (7). This equation is similar to Eq. (6), in that it calculates the proportion of material exchange by bioturbation for a certain donor layer by integrating and dividing exponential depth functions. Eq. (7) integrates an exponential equation over the vertical distance from the centre depth of the donor layer ($z_{layer}$) to the upper and lower depths of an exchange layer ($z_{upper}$, $z_{lower}$). This integral is divided by the sum of the integrals of two other exponential equations, starting from $z_{layer}$ and going towards the soil surface (z = 0) and towards the bottom of the soil profile (*sd*). Through this equation, the amount of exchange between a donor layer and an exchange layer decreases with increasing distance between the layers, leading to diffusive mixing. The gradient of these exponential equations is controlled by depth parameter $dd_{mix}$ [m$^{-1}$].

$$BT_{exchange} = BT_{layer} * \frac{\left| \int_{z_{layer}-z_{upp}}^{z_{layer}-z_{lower}} e^{(-dd_{mix}*z)} dz \right|}{\int_0^{sd-z_{layer}} e^{(-dd_{mix}*z)} dz + \int_0^{z_{layer}} e^{(-dd_{mix}*z)} dz} \tag{7}$$

### 2.3.4    Model set-up

The model requires several parameters as input. These parameters can be grouped in environmental parameters that are determined by the organisms, ecosystem and climate (type of mixing processes, depth of active mixing zone, bioturbation period), model-based parameters that determine the configuration and construction of the modelled soil (soil and layer properties, bleaching depth) and process-based parameters that control process behaviour (bioturbation rate, depth functions and their parameters, combination of processes). We ran our simulations using the values in Table 2 to illustrate how bioturbation affects luminescence-based depth profiles. These parameters should be constrained with experimental data or through inverse modelling when applied to real-world settings.

The environmental and model-based parameters were the same for all the simulations. We ran our simulations in a one-dimensional soil profile (pedon), to focus on vertical mixing processes and avoid effects from lateral redistribution processes. We simulated bioturbation over a period of 10 ka with an annual time step and with an active mixing zone of 1 m deep. Due to the diffusive transport of subsurface mixing, material sourced in the active mixing zone can also be transported to layers below the active mixing zone. To account for this effect, we perform the simulations on a 2 m deep pedon. The pedon contains 200 soil layers of 1 cm thick, with an upper layer of 5 mm representing the bleaching depth. The bleaching depth is based on model-based estimates (Furbish et al., 2018b) and is in line with light penetration depths in rocks (0-15 mm, Meyer et al.,

2018). Each layer initially contains 150 luminescence particles. We simulate a uniform loess-like soil texture (25% sand, 60% silt, 15% clay) with a constant bulk density of 1500 kg m$^{-3}$ to avoid effects of textural and density variations on the age

distributions in the simulations.

To study how the different processes and their parameters can influence luminescence-based depth profiles, we adjusted the process-based parameters in turns according to the values reported in Table 2 under scenario variations. The standard total bioturbation $BT_{pot}$ was set to 10 kg m$^{-2}$ a$^{-1}$ (loosely based on rates reported in Wilkinson et al., 2009: 0.3-110 kg m$^{-2}$ a$^{-1}$) and was varied from 1 – 10 kg m$^{-2}$ a$^{-1}$. The standard depth function was gradational, but was also varied with exponential and

210 abrupt depth profiles. The depth parameters were selected such that the active mixing zone is restricted to the upper 1 meter of the pedon to facilitate comparison between the simulations (Figure 2). The two bioturbation processes were combined with various contributions, ranging between 0 and 100%.

**Table 2: The parameters used in this study, categorized by type. The reported values remained constant throughout the simulations, except when adjusted according to the scenario variations listed in the last column.**

| Parameter type | Description | Value | Scenario variations |
|---|---|---|---|
| Environmental | Process | Mounding, subsurface mixing | |
| | Depth mixing zone | 1 | |
| | Bioturbation period [ka] | 10 | |
| Model-based | Soil depth [m] | 2 | |
| | Number of layers | 200 | |
| | Initial layer thickness [m] | 0.01 | |
| | Number of grains per layer | 150 | |
| | Bleaching depth [m] | 0.005 | |
| | Soil texture [% sand, silt, clay] | 25, 60, 15 | |
| | Bulk density [kg m$^{-3}$] | 1500 | |
| Process-based | Total bioturbation $BT_{pot}$ [kg m$^{-2}$ a$^{-1}$] | 10 | 1 – 10 |
| | Depth function $df$ with depth parameter $dd$ [m$^{-1}$] in brackets | Gradational (1) | Gradational (1), Exponential (6), Abrupt (1) |
| | Exchange parameter $dd_{mix}$ [m$^{-1}$] | 10 | |
| | Relative contribution of processes [%] | 100 | 0-100 |

**2.4    Data presentation and comparison**

The model produces a large amount of data, as there are multiple simulations with a large number of layers that all contain about 150 luminescence particles. To facilitate visualization and comparison between the different model scenarios, we took

two steps to summarize the model output before presentation. First, we aggregated the model output per five layers, so that their thickness resembles typical 5-cm thick OSL samples. This reduced the scatter resulting from the stochastic particle transport. Second, we present the simulated ages as age distributions using probability functions (Bateman et al., 2003), which we then summarized with different metrics. Working with age distributions instead of statistical age models (e.g. Galbraith and Roberts, 2012) provides the advantage that we don't need to select a suitable age model and estimate its corresponding statistical parameters for each individual sample. This allows us to automate and speed up the modelling and calibration process (see Section 4.3), without introducing uncertainties from age model selection. A disadvantage of this approach is that we don't get a robust estimate of the error of the estimated age, but that is not required in this study. Because we expect non-normal or even multimodal distributions in the data, we calculated the probability functions using a bandwidth following the method of Sheather and Jones (1991), which was developed for non-normal distributions. Saturated or non-bleached grains were excluded from the probability functions. We use the following metrics to summarize the age distributions:

- The modal age, which corresponds to the highest peak in the age distribution, which we consider the most likely burial age of the sample or layer;
- The interquartile range, as a robust measure of the width of the distribution;
- The bioturbated fraction($f_{bio}$, Eq. (1)), as a measure of the fraction of bleached particles due to bioturbation.

Detailed plots of the simulated age distributions are provided in Supplementary Figures S1-S3.

For the comparison of experimental data and simulations, we normalized the depths and luminescence ages. For the experimental data, we normalized depths by dividing sampling depth by the maximum sampling depth and the luminescence ages by dividing the individual grain ages by the extent of the bioturbation period or the saturation criteria (Table 1). For the simulations, the depth was normalized by dividing simulation depth by the active mixing depth of 1 m. The simulated ages were normalized by dividing by the simulation time of 10 ka.

## 3    Results

### 3.1    Experimental studies

Figure 3 shows the luminescence-based depth profiles from the experimental datasets. The plots are in order of increasing contribution of subsurface mixing (termites -> worms -> ants). With a larger contribution of subsurface mixing, the bioturbated fractions in the subsurface decrease. The termites and worms datasets show clear age-depth trends, while the ants dataset shows a more scattered depth profile with a discontinuity in the modes. The termites and ants datasets show an increasing interquartile range with depth, while the worms dataset shows relatively constant interquartile range. There are also clear differences in the bioturbated fraction. The termites dataset has a bioturbated fraction over 50% for the entire profile with over 90% bleaching in the upper 60 cm. The worms dataset also has well-bleached upper samples, but the bioturbated fraction approaches 25% for the lowest sample. For the ants, only the upper sample shows good bleaching, with a bioturbated fraction of 97%. This drops to 12% and 6% deeper in the profile, where only 6 to 8 samples contain a luminescence signal.

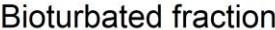

**Figure 3: Age-depth profiles for the experimental datasets used in this study: (a) termites (Kristensen et al., 2015), (b) anecic earthworms (Von Suchodoletz et al., 2023), (c) ants (Román-Sánchez et al., 2019a). The bottom axes show the ages of the measurements. The upper axes show the bioturbated fraction. Where provided, the red dashed line indicates the period of bioturbation by the current agent (Table 1).**

## 3.2 Comparison of depth functions and bioturbation rates

The simulations of separate mounding and subsurface mixing processes with varying depth functions show clear differences in the resulting depth profiles (Figure 4). The mounding shows curved age-depth trends with low interquartile ranges for all different depth functions, which slightly increase closer to the lower boundary of the active mixing zone of 1 m (Figure 4a). The gradational and exponential profiles approach the simulation time of 10 ka at the bottom of the profile, while the abrupt

profile has much younger ages and a steeper depth profile. For each depth profile, almost all particles have been bioturbated and bleached in the active mixing zone, as shown by the bioturbated fraction. Below this zone, none of the particles are bioturbated.

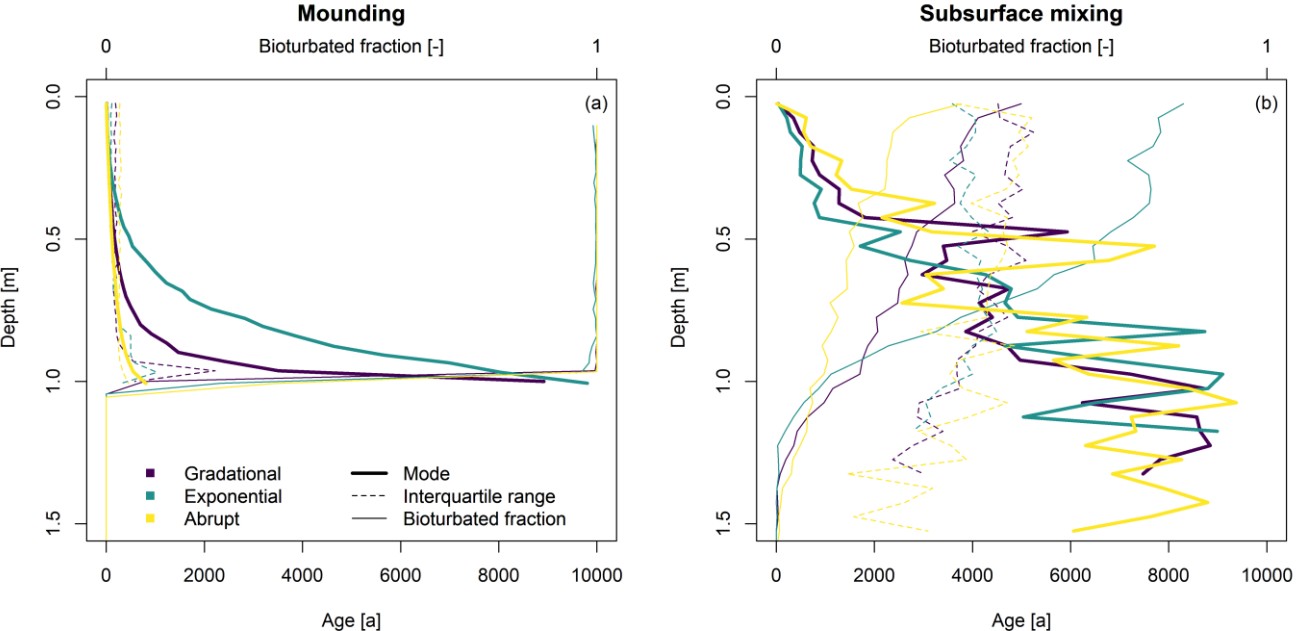

**Figure 4: Luminescence-based depth profiles resulting from simulations of (a) mounding and (b) subsurface mixing, using different**
**depth functions, with potential bioturbation rates of 10 kg m⁻² a⁻¹. Detailed plots of the simulated luminescence ages and their distributions are provided in Fig. S1.**

In contrast, the simulations of subsurface mixing show more chaotic, heterogeneous depth profiles of modal age, that only show a general increasing trend with depth (Figure 4b). The scatter in the depth trends of the modal ages also increases with depth and even reaches below the active mixing zone of 1 m, due to exchange of material from bioturbated layers with all
270 other soil layers. It should be noted that below 1 meter there are only a few bioturbated grains present. The modes are similar for each simulated depth function, with slightly lower interquartile ranges for the upper part of the exponential depth profile. The interquartile ranges are high for all simulations, and generally decrease down the profile. This concerns only a small number of particles, as evidenced by the bioturbated fraction. The amount of bioturbated particles decreases with soil depth. The exponential profile contains most bioturbated particles, followed by the gradational and abrupt profiles.

Variations in the bioturbation rate for the mounding and subsurface mixing processes shows a clear effect on the steepness of the age-depth curves (Figure 5). For the mounding process, higher rates lead to a steeper age-depth profile. Throughout the bioturbated profiles, almost all luminescence particles have been bleached, independent of the rate. For the subsurface mixing process, higher rates show younger modal ages and higher bioturbated fractions. The interquartile ranges show comparable trends, with different levels of scatter in the depth trends. Bioturbation rates also affect the depth of the mixed profiles, where
lower bioturbation rates lead to shallower mixing bioturbated profiles.

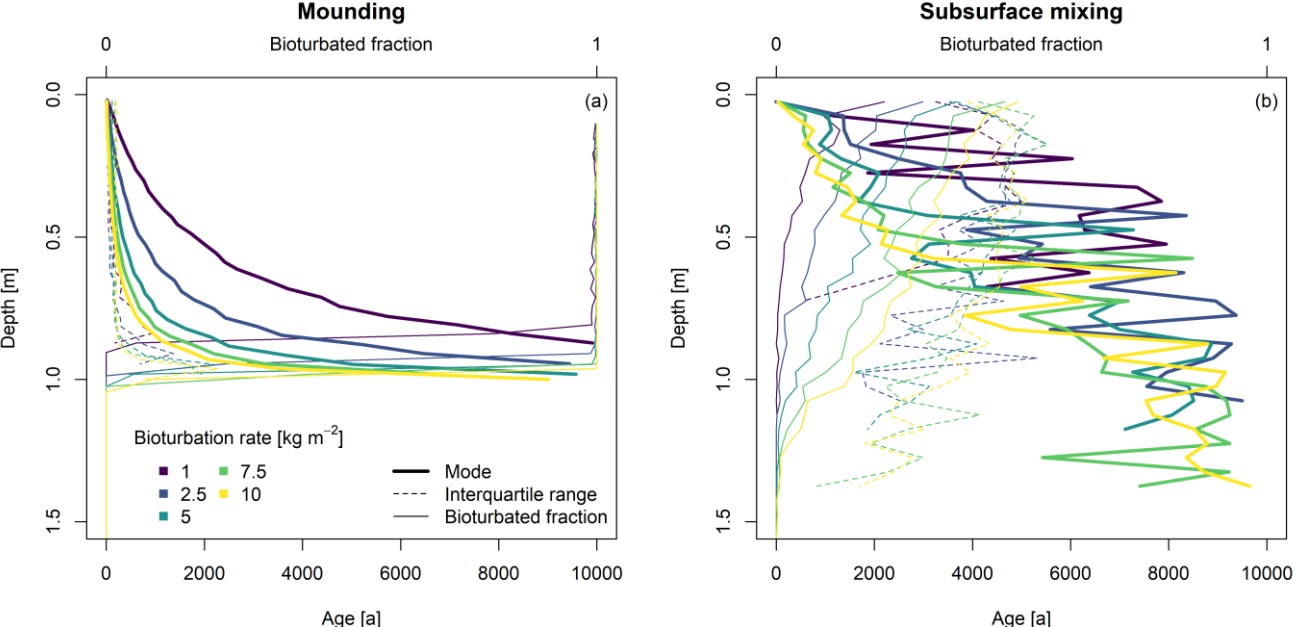

**Figure 5: Age-depth profiles resulting from bioturbation by (a) mounding and (b) subsurface mixing, using a gradational depth profile and varying bioturbation rates. Detailed plots of the simulated luminescence ages and their distributions are provided in Fig. S2.**

### 3.3 Combination of mounding and subsurface mixing

Simulations where mounding and mixing were combined in different ratios show that the mounding process dominates the age-depth characteristics (Figure 6a). Only when the fraction of mounding decreases to less than 5%, the depth curves start to resemble the profile with solely subsurface mixing. The same pattern is visible for the bioturbated fraction, but the interquartile range reacts quicker to changes in the ratio of mounding and subsurface mixing. Overall, a larger contribution of subsurface mixing leads to older luminescence particles in the profile (Figure 6a), wider age distributions (Figure 6b) and less bioturbated particles (Figure 6c).

The general trends in the experimental data conform with the trends in the simulation data. Termites, as mounding organisms, show lower modes of ages compared to worms, which both mound and mix (Figure 6a). The patterns in the topsoil are similar for all simulations and experimental datasets, but in the subsurface the termites, as mounding organisms, show lower modes of ages compared to worms, which both mound and mix (Figure 6a). The ants dataset shows lower modal ages than both other organisms. This can be attributed to the normalization procedure: while the observed period of bioturbation had been constrained in case of the termites and worms dataset, the ants dataset had no such constraint and was consequently normalized by much higher age values, leading to lower normalized ages (Table 1). The worms dataset shows higher interquartile ranges compared to the termites dataset. (Figure 6b). Also in this case, the ants dataset forms the exception due to the high normalization age. The simulated interquartile ranges are much lower for mounding-dominated scenarios than the experimental

studies, indicating an underestimation of the spread in age distributions. The bioturbated fraction also shows clear differences between mounding and subsurface mixing organisms, with a higher proportion of bioturbated grains for mounding organisms and mounding simulations (Figure 6c).

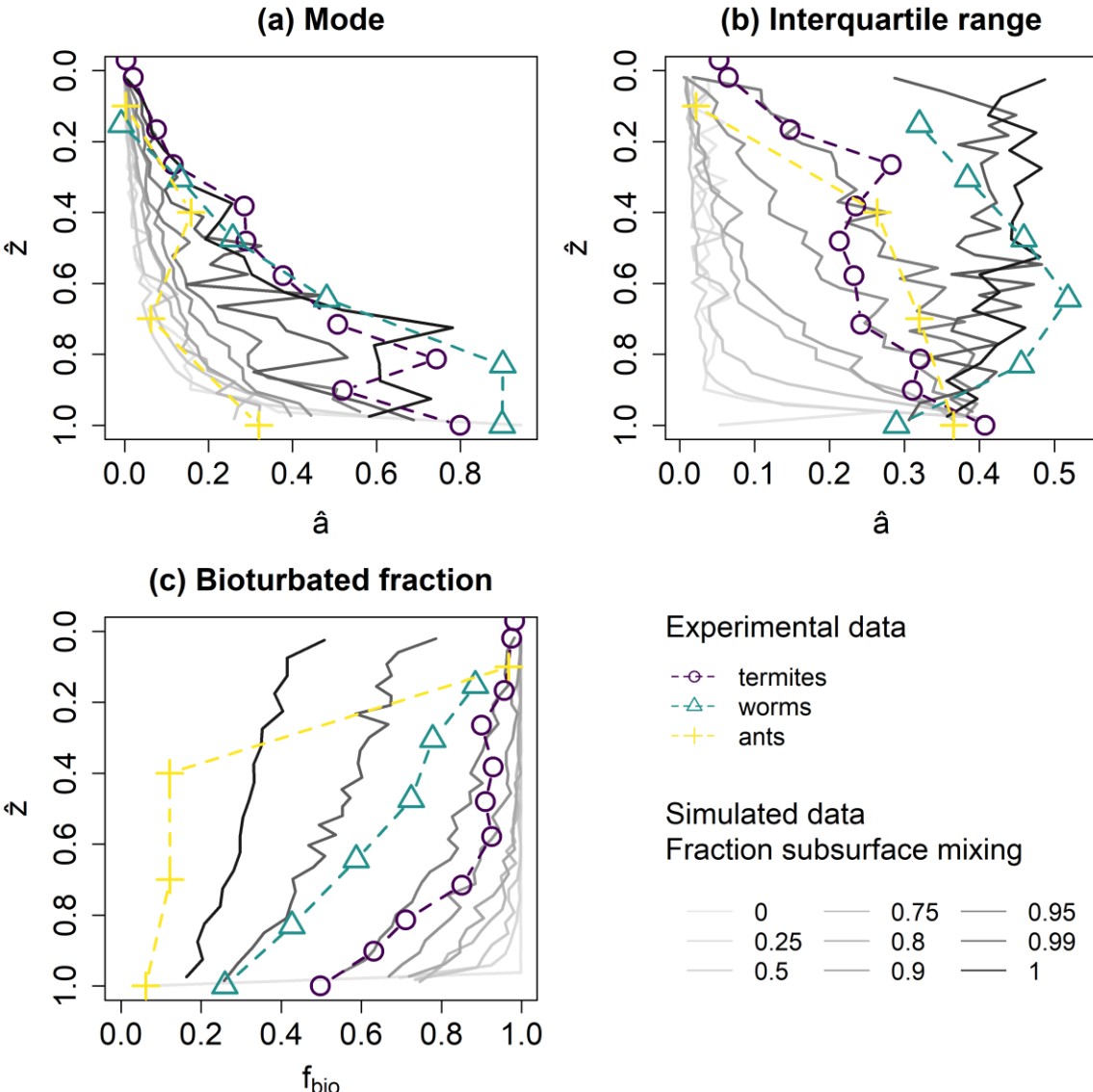

**Figure 6: Statistics for mixes between mounding and subsurface mixing (grey lines), aggregated per 5 soil layers (~ 5 cm), compared to the experimental datasets (coloured lines and points). Results were normalized for age (â) and depth (ẑ). The different windows show different statistics: (a) mode of age distributions; (b) interquartile range; (c) bioturbated fraction $f_{bio}$. The simulations were run with a gradational depth function and active mixing zone of 1 m, with a total bioturbation rate of 10 kg m$^{-2}$ a$^{-1}$, divided over the two processes. Detailed plots of the simulated luminescence ages and their distributions in this plot are provided in Fig. S3.**

## 4    Discussion

### 4.1    Mixing patterns by mounding and subsurface mixing

Bioturbation induces different mixing patterns in the soil, depending on the organism and process. By the integration of luminescence tracers and numerical simulations, we identified distinct ways in which different processes and parameters impact soil mixing. Here, we will elaborate on the processes, their effects on luminescence tracers and show that they are consistent across experimental datasets.

The upward advective transport of soil material by mounding animals continuously buries previously mounded material, which leads to age-depth profiles in the active mixing zone that resemble depositional profiles. The continuous upward transport of material to the surface results in a high degree of bleaching and consequently narrow age distributions, as evidenced by the termites dataset and the numerical simulations (Figure 3a; Figure 4a). The lower boundary of the active mixing zone is often characterized by an abrupt increase in ages, changing widths of age distributions, lower age-depth rates and a decrease in the bioturbated fraction. This is clearly visible in the termites study by Kristensen et al. (2015), where the fraction of saturated grains increases from 0–4 % in the active mixing zone to up to 60% in the layers below (data not shown), accompanied by a jump in the luminescence ages and increase in the uncertainties. The same is visible in the data from Madsen et al. (2011), who measured luminescence-based age-depth curves from aliquots collected in tidal flats, which are bioturbated by mounding lugworms. There is a clear distinction between the active mixing zone, with narrow age distributions and steeper age-depth gradients, and the underlying depositional sequence.

The age-depth curves of bioturbation by subsurface mixing display completely different characteristics (Figure 6). The limited bleaching at the surface and diffusion-like transport leads to a low population of bleached particles in the subsurface and wide luminescence distributions. The stochastic nature of particle transport by subsurface mixing is clearly visible in the ants dataset (Figure 3c; Román-Sánchez et al., 2019a), with only a few luminescent grains in the subsoil that show a high age range. Ants often create mounds at their nest entrances (Richards, 2009), suggesting that luminescence-based depth profiles for ants should contain mounding signals as well. Román-Sánchez et al. (2019a) studied a profile on a hilltop with an equilibrium between soil erosion and soil production. If the erosion primarily removed the surface mounds, the subsurface mixing component of bioturbation would be amplified. The low bioturbated fraction and wide age distributions are also consistent in other luminescence datasets with considerable subsurface mixing components, for example by root activity (Stockmann et al., 2013; Johnson et al., 2014), or sites where mounded material by ants and wombats has been washed away by overland flow (Heimsath et al., 2002; Wackett et al., 2018). Two of these datasets only contained a small proportion of non-saturated grains (Heimsath et al., 2002; Stockmann et al., 2013). Surprisingly, the data of Johnson et al. (2014) had a very low number of saturated grains in their dataset, which they attribute to an aeolian input of bleached quartz grains. Erosion by water or soil creep can result in shallower bioturbated profiles with older ages varying impacts on bioturbated fractions. Water erosion tends to produce lower bioturbated fractions, while soil creep leads to higher bioturbated fractions (Román-Sánchez et al., 2019a). These effects are comparable to those caused by changes is bioturbation rates in stable landscape positions. Therefore, it is important to consider

the potential occurrence of erosion before interpreting luminescence-depth profiles resulting from bioturbation, as it can substantially change the interpretation. The worms and termites datasets were collected from flat terrain and were therefore not significantly impacted by erosion processes.

In addition to the studied processes, there are various other forms of bioturbation. One example is upheaval, involving the sudden detachment, homogenization, and re-deposition of soil. For example, when a tree is uprooted, the soil from the root clump falls back into the pit (Gabet et al., 2003). Ploughing could also be considered upheaval. Here, a body of soil is efficiently detached, turned over and redeposited, for example by a mouldboard plough (De Alba et al., 2004; Van der Meij et al., 2019). Due to its constant mixing rate with depth and homogenization of the soil, upheaval likely produces relatively homogeneous age distributions in the active mixing zone, with an abrupt increase in age below this zone. The ages, distribution widths and bioturbated fractions depend on the frequency and mixing depth of upheaval. We expect that upheaval did not contribute to the experimental datasets used in this study, as they were sampled from sparsely forested savannah ecosystems and there are no homogeneous age distributions in the active mixing zones (Figure 3). Bioturbation by upheaval, and its interactions with mounding and subsurface mixing, will be explored in future research.

The distinct effects of different bioturbation processes on soil fluxes that we identified here emphasize the necessity of including multiple formulations of bioturbation processes in soil evolution models and soil function models, as conventional diffusion-type subsurface mixing processes account for only a part of soil mixing.

## 4.2    Luminescence as tracer of soil mixing processes

Luminescence-based tracers rely on the exposure and bleaching of soil particles to daylight at the surface. Bleached particles are transported downward by various processes, where they can be used as a tracer for soil mixing. As a result, luminescence primarily traces downward transport within soils (Gliganic et al., 2016). Luminescence-based age-depth profiles are predominantly influenced by mounding, because this process exposes more grains to daylight, and therefore do not adequately represent subsurface mixing processes (Figure 6a). The interquartile range, as proxy for the width of the age distributions, reacts quicker to changes in the balance of mounding and subsurface mixing (Figure 6b). This suggests that the interquartile range might be key in separating between mounding and subsurface mixing signals using luminescence-based tracers, which is still one of the main challenges of determining bioturbation rates (Wilkinson et al., 2009; Halfen and Hasiotis, 2010). This will be explored further in Sect. 4.3.

The bioturbated fraction acts as a downward tracer of soil mixing due to supply of bleached grains from the surface, but can act as an upward tracer of soil mixing as well (Reimann et al., 2017). Bedrock weathering, increased bioturbation or surface denudation lead to the to the downward migration of the active mixing zone, introducing saturated grains from the bottom up into the soil profile. These processes were not accounted for in this study. However, they do play a significant role in interactions between bioturbation and hillslope processes (Román-Sánchez et al., 2019b, a) and should be taken into account when applying bioturbation models in two to three-dimensional settings.

The modal ages, interquartile range and bioturbated fraction are not only influenced by the type of bioturbation processes, but also by the applied depth function and process parameters such as soil mixing rate. In Figure 7, we have compiled an overview of how different processes, implementations and parameters affect the depth functions of luminescence-based metrics. The characteristics of these depth functions offer qualitative insights into the characteristics of the underlying mixing processes.

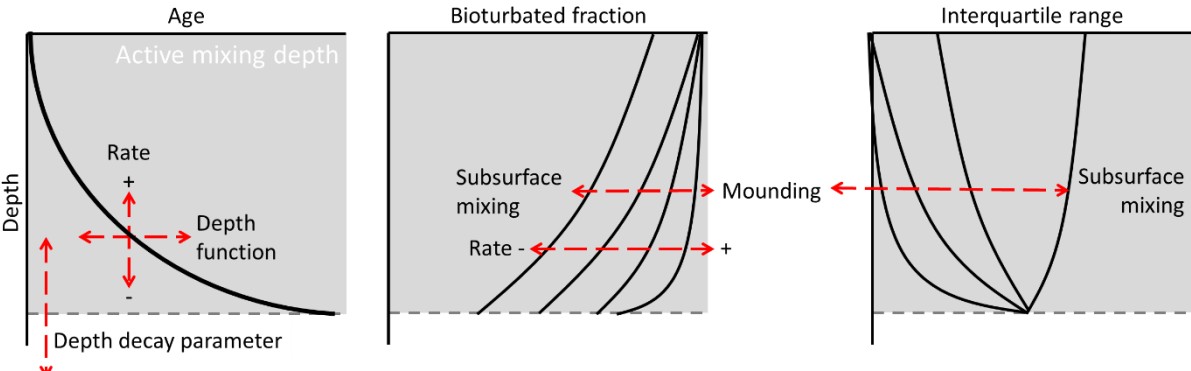

**Figure 7: Conceptual overview showing how different factors and parameters affect the depth profiles of luminescence-based metrics.**

    The majority of soil mixing happens in the active mixing zone, with the maximum depth being determined by the reach of organisms into the soil (Figure 3), represented by the depth decay parameters in the simulations. This zone is distinguished from underlying layers by younger, measurable ages and a higher bioturbated fraction. It is challenging to determine the depth

function of mixing processes from age-depth profiles. This supports earlier statements about determining depth dependency of soil mixing (Gray et al., 2020). The steepness of the exponential age-depth profiles can either be a result of a different depth function or a different soil mixing rate. The dominant mixing process can be derived from the bioturbated fraction and the interquartile range, where a higher proportion of mounding results in higher bioturbated fractions and lower interquartile ranges.

The combination of luminescence-based ages, the interquartile range and the bioturbated fraction provides a comprehensive toolbox for tracing soil mixing processes. Ideally, these tracers are combined and verified with independent tracers that trace either downward or upward transport. Fallout radionuclides or meteoric cosmogenic radionuclides are examples of downward-oriented tracers (Tyler et al., 2001; Kaste et al., 2007; Johnson et al., 2014), while in situ created cosmogenic nuclides (Heimsath et al., 1997; Brown et al., 2003) and reworked clay coatings originating from Bt horizons (papules, Miedema and

Slager, 1972; Sauzet et al., 2023) are produced in or below the soil column and therefore can act as upward-oriented tracers. Numerical methods such as ChronoLorica provides a flexible platform to integrate different soil mixing tracers and simulate their distribution in complex multi-mixed environments.

### 4.3 Towards a quantitative evaluation of luminescence-based depth profiles

This qualitative understanding of the luminescence-based depth profiles, coupled with a model capable of simulating various bioturbation processes, sets the stage for a quantitatively determining the impact and rates of different bioturbation processes through model calibration. Here we make a first attempt to showcase the potential of the model to derive quantitative bioturbation parameters through calibration. We do this for the termites and worms datasets, using the accompanying model Mixed Signals (See Sect. 4.4). We do not attempt a calibration for the ants dataset, because the effects of erosion and soil formation on this profile are not sufficiently constrained in the model.

For the calibration, we follow the same categories of parameters as reported in Table 1. We based the environmental parameters on field observations and experimental results. We used the same model-based parameters as reported in Table 1, with the exception of the layer thickness. This was set to 2 cm to increase calculation speed. At this stage, the bioturbation is not grain-size specific, so the model output is insensitive to differences in parent material composition. Therefore, these were not modified for the calibration.

To determine the process-based parameters, we ran the model with varying depth functions, potential bioturbation rates and contributions of mounding and subsurface mixing. We determined the parameter set that produced the closest match with the experimental data by minimizing the combined squared error ($error_{squared}$) of experimental and simulated modal age, interquartile range and bioturbated fraction (Eq. (8)), where $P$ is the number of luminescence metrics and $O$ is the number of observations in the experimental dataset.

$$error_{squared} = \sum_{p=1}^{P} \sum_{o=1}^{O} \left( p(o_{simulated}) - p(o_{experimental}) \right)^2 \qquad (8)$$

Calibration across all three metrics enabled us to capture the majority of the dynamics observed in the depth profiles resulting from different processes and parameters (Figure 7). To ensure equal weighting of the three metrics, the ages were normalized by dividing them by the runtime (i.e. bioturbation period) of the model. Consequently, all metrics have potential values ranging from 0 to 1. Alternatively, the evaluation metric could be based on statistical tests that measure the similarity between the experimental and simulated age distributions, such as the Kolmogorov-Smirnov Test or Earth Mover's Distance.

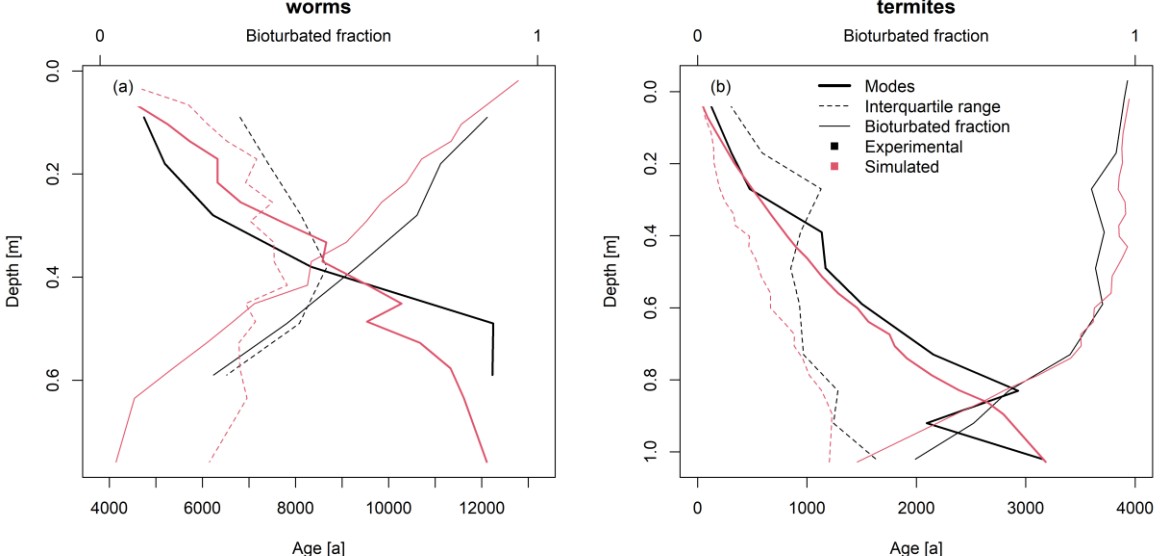

**Figure 8: Calibration results for the (a) worms and (b) termites datasets. Initial layer thicknesses in the model were 2 cm. To reduce scatter in the visualization of the model results stemming from the stochastic particle transport process, the simulated results (in red) are aggregated per three layers, resembling typical 5-cm thick OSL samples.**

The model is well equipped to reproduce the experimental luminescence-based depth profiles (Figure 8). The simulated depth profiles of the three metrics approach the experimental depth profiles, with some deviations due to fluctuations in the experimental data and the calibration on three different metrics. For the worms dataset, the parameter set that resulted in the lowest squared error was a gradational depth profile, potential bioturbation rate of 1.5 kg m$^{-2}$ a$^{-1}$, 90% subsurface mixing and 10% mounding. This ratio of processes agrees well with our expectations for burrowing anecic earthworms, which mainly live

underground and sometimes visit the surface (Taylor et al., 2019). The parameter set that resulted in the lowest squared error for the termites dataset was an abrupt depth profile with a bioturbation rate of 4.5 kg m$^{-2}$ a$^{-1}$, with 80% subsurface mixing and 20% mounding. We expected a much higher contribution of mounding for the termites due to their construction of large surface mounds. However, a component of subsurface mixing was also expected, as termites transport material in the subsurface when they mine material for their mounds, similar to ant subsurface galleries (Rink et al., 2013). The abrupt depth profile that was

calibrated for the termites data contradicts the findings of Gray et al. (2020), who found that mixing rates generally decrease with depth.

Interestingly, the calibrated bioturbation rates are multiple orders of magnitude larger than the soil reworking rates reported in the original studies (~40 g m$^{-2}$ a$^{-1}$ for termites, Kristensen et al., 2015; ~20 - 80 g m$^{-2}$ a$^{-1}$ for worms, von Suchodoletz et al., 2023). These reported rates were based on measured OSL ages and their depths. These ages represent the current burial ages

of the grains, but do not account for previous resurfacing of grains or subsurface transport without bleaching. Hence, they represent only the net displacement of soil particles from the surface to the subsurface. The calibrated rates are in the same order of magnitude as rates of mounding and mixing determined by earthworm ingestion rates and weighing worm casts and

surface mounds (see compilation in Wilkinson et al., 2009). Based on these factors, the actual bioturbation rates in the studied sites are probably closer to the calibrated rates than to the OSL-based soil reworking rates.

This modelling exercise provides unique opportunities to quantitatively distinguish mounding and subsurface mixing processes. However, the current results do not match with our expectations, especially for the termites dataset. This discrepancy is probably a consequence of the assumption of complete bleaching within the bleaching depth in the model. The bleaching depth of 5 mm in this study was based on model-based estimates (Furbish et al., 2018b) and is in line with light penetration depths in rocks (0-15 mm, Meyer et al., 2018). However, in reality, not all near-surface grains are bleached, due to the

attenuation of light after it penetrates the soil surface and the formation of soil aggregates, which shield inner particles from light. Notably, the agents responsible for soil mixing are also largely responsible for soil aggregation (Lee and Foster, 1991; Bottinelli et al., 2015). A lower bleaching efficiency – the fraction of particles that is bleached within the bleaching depth – would result in lower bioturbated fractions and higher interquartile ranges, which are the same effects that a larger contribution of subsurface mixing would have.

The bleaching depth and bleaching efficiency need to be better constrained before accurate calibration of the experimental profiles is possible. These model-based parameters could be estimated through model calibration, but this comes with the risk that multiple parameter combinations could result in equally plausible mixing scenarios, as bleaching efficiency and subsurface mixing have similar effects on the calibration parameters. Experimental evidence on bleaching depths and bleaching efficiency in soils, which likely vary across soil types and vegetation cover, is thus required to constrain these parameters and provide

accurate, quantitative estimates of bioturbation rates and processes based on luminescence tracers and numerical modelling.

### 4.4    Simulation tool for bioturbation

The simulations presented in this paper were modelled with ChronoLorica, which is a comprehensive soil-landscape evolution model that simulates multiple pedogenic and geomorphic processes, together with multiple geochronometers (Van der Meij et al., 2023). The model, without the new formulations for bioturbation, is available via the Zenodo repository (Van der Meij and

Temme, 2022).

We also developed a separate model, named Mixed Signals, which contains the formulations of bioturbation processes and their effects on luminescence tracers, as described in this paper, as well as visualization and calibration tools. This model can be used or adapted for simulating bioturbation effects on luminescence-based tracers, for example in explorative studies or for education purposes. The model is written in Julia, which is an interactive high-performance scientific computing language

(Bezanson et al., 2017). The Mixed Signals model is freely available https://github.com/MarijnvanderMeij/Mixed-signals_Bioturbation and will be published to the Zenodo repository after review of this paper. The download contains the following files:

- a readme file with instructions to launch the model;
- a Jupyter Notebook with illustrative examples demonstrating how to use the model to simulate soil mixing and its
effects on luminescence-based depth profiles;

- a script with all the functions that are required to run the model and create visualizations;
- a synthetic luminescence dataset for illustrating the calibration process.

## 5    Conclusions

Soil bioturbation plays a crucial role in soil functions and soil evolution by cycling carbon and nutrients, but there is limited
knowledge on how different mixing processes affect fluxes and rates of soil material. In this study, we combined experimental
luminescence-based datasets and numerical modelling to study two main bioturbation processes – mounding and subsurface
mixing – and their respective mixing patterns. These mixing patterns have distinct effects on luminescence tracers, which we
characterized with three metrics: the modal age of the age distribution as most probable burial age of each layer, the
interquartile range as measure of the width of the distributions and the bioturbated fraction as the fraction of bleached particles
in each layer.

By numerically simulating mounding and subsurface mixing with varying rates, depth functions and interactions between
processes, we determined how each process affects the luminescence-based depth profiles. Mounding is an advective process
that moves soil material to the surface, leading to a high degree of luminescence signal resetting (bleaching), low interquartile
ranges and a high bioturbated fraction. Subsurface mixing is a diffusive process, which transports a much lower number of
grains from the surface, leading to high interquartile ranges and low bioturbated fractions. We summarized these effects in a
conceptual diagram to facilitate qualitative interpretation of luminescence-based depth profiles.

A first attempt to quantitatively interpret luminescence-based depth profiles through model calibration showed that the model
is able to reproduce the experimental depth profiles and provide realistic bioturbation rates. The model is not yet equipped to
accurately determine the relative contribution of mounding and subsurface mixing in the experimental datasets, likely due to
overestimating the degree of bleaching at the surface. Experimental data on bleaching depth and bleaching efficiency in soils
is required before accurate, quantitative estimates of bioturbation rates and processes can be determined.

Our compilation of luminescence-based soil tracer studies and numerical simulations shows that bioturbation is more than a
simple diffusive mixing process. Different organisms cause different transport processes in the soil, with major differences in
fluxes of soil material and consequently nutrients and carbon. We provide numerical formulations of two main bioturbation
processes, which could be used to improve soil function and soil evolution models. The accompanying model Mixed Signals
contains these implementations and can be used for explorative studies, education purposes and quantitative determination of
bioturbation parameters through model calibration.

## Code and data availability

The luminescence data used in this study are published in earlier work (Kristensen et al., 2015; Román-Sánchez et al., 2019a;
Von Suchodoletz et al., 2023) and we refer to the authors of these works for data requests. The ChronoLorica model is publicly

available via https://doi.org/10.5281/zenodo.7875033 (Van der Meij and Temme, 2022). The new bioturbation implementations can be found in the maintained versions of ChronoLorica and other versions of Lorica through https://github.com/arnaudtemme/lorica_all_versions (last access: 11 September 2024), and will be added to a new version of the model. The model Mixed Signals is available via https://github.com/MarijnvanderMeij/Mixed-signals_Bioturbation and will be published in the Zenodo repository after review of this paper.

**Author contributions**

**W. Marijn van der Meij**: Conceptualization, Methodology, Software, Visualization, Writing – Original draft. **Svenja Riedesel**: Conceptualization, Investigation, Writing – Review & Editing. **Tony Reimann**: Conceptualization, Writing – Review & Editing.

**Competing interests**

The authors declare that they have no conflicts of interest.

**Acknowledgements**

We thank Jeppe Aagaard Kristensen (Aarhus University) and Kristina Jørkov Thomsen (Technical University of Denmark) for sharing the termites data and Andrea Román-Sánchez for sample preparation and measurements of the ants data. Hans von Suchodoletz (University of Leipzig) is thanked for collecting samples and providing context information for the worms data. We are also grateful to the reviewers and editors for their valuable feedback on the manuscript. SR acknowledges support by the European Union's Horizon Europe research and innovation programme (RECREATE, grant no. 101103587) during co-writing and editing of the manuscript. The Open Access publication costs were supported by the Deutsche Forschungsgemeinschaft (DFG, 491454339).

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
