# Peer review of "Mixed signals: interpreting mixing patterns of different soil bioturbation processes through luminescence and numerical modelling"

_EGUsphere, 2024_

## Author Response (AR1)

**Author's response**

Dear Katerina Georgiou,

Thank you for handling our submission to SOIL. We have now finished the revised manuscript, where we incorporated the referees' comments based on our rebuttals to their respective reviews. We attached a copy of these rebuttals below. Here we list the major changes to the manuscript based on the major comments of the reviewers:

- Equations 3-5: We revised the depth function equations and descriptions to correct a previous error.
- Section 2.3.4: We added a more extensive description of model parameters and listed them in a new version of Table 2, which now includes the model parameters as well as their variations for simulating different scenarios.
- Section 2.3.5: We better motivated how we post-processed the simulated data and provided a stronger motivation for our choice of the used statistics to visualize these results.
- Supplementary information: We added a SI that provides detailed plots of simulated ages and their distributions for all presented simulation results. This document contains 25 sub-plots, making it impossible to provide this information in the manuscript itself.
- References: we added four relevant references, of which one was suggested by referee 2 and two were mentioned by referee 3 in his review.
- General: we added many small clarifications to improve the readability of the text and the reproducibility of the work.

We want to thank all reviewers for their valuable comments, as we believe they have made the manuscript more understandable and hopefully suitable for publication. One last thing that we need to do is to publish the accompanying model code on the Zenodo repository and provide a reference to this in the manuscript. I would like to do this when the DOI of the manuscript is available, so that we can use that in the model README to refer to the accompanying manuscript.

With best regards,

On behalf of all authors,

Marijn van der Meij

**Response referee 1**

Dear Shannon Mahan,

Thank you for your kind words and positive review. We will take your comments into account in the new version of the manuscript. See our detailed responses below.

On behalf of all authors,

Marijn van der Meij

*GENERAL COMMENTS:*

*The authors have done a commendable job is taking a system that is very complex and reducing it understandable and concise sections with their ChronoLorica modeling. I could not find much to quibble about with the equations and modeling. Sometimes I was unclear about a parameter or decision but overall in the paper I found the murkiness resolved itself upon further reading. The following comments are meant to help refine small points I found need either editing, additional clarification, or deleting.*

*Overall the conclusions are justified given the discussion and often confusing scatter in the measurements and model data. I look forward to the next round that includes ants and one site with both quartz and feldspar.*

*SPECIFIC COMMENTS:*

*There is an over-use of the word "fundamental in the first paragraph of the introduction. I think that the word "fundamental" is perhaps not quite the correct word to describe what is lacking about our knowledge of soil processes or the way the organisms bioturbate. We actually know a lot about the fundamental details of soil (see the endless journals devoted to it such as Soils Systems, European Journal of Soil Science, Soil Advances, etc.) we just don't know the depth per dominate organism or the depth is highly regional. We also don't whether mounding or subsurface dominates at any given depth (some being obvious in the top 10 cm). I think the words you are looking for are a "simple tracer" to measure depth of organism effect on soil processes. You actually say it in lines 65. Rewrite to indicate the "fundamental" question is not about soil processes but the depth and type of processes per a known biological agent.*

**Response***:* Thank you for the clarification.  We will rephrase this paragraph to: "Bioturbation is the umbrella term for soil mixing processes by various organisms. Bioturbation plays a key role in soil nutrient cycling, carbon sequestration, erosion, and the redistribution of contaminants and pollutants (Wilkinson et al., 2009; Briones, 2014; Creamer et al., 2022). Despite its pivotal role in regulating soil functions, we have an incomplete understanding regarding how different organisms and ecosystems impact the types and rates of mixing processes, how these rates vary with soil depth and how different mixing processes interact within the soil (Schiffers et al., 2011; Michel et al., 2022). These insights are essential for accurately modelling the effects of bioturbation on present-day soil functions and the long-term evolution of soils (Creamer et al., 2022; Meng et al., 2022)."

*Table 1. I find it interesting that you used quartz for the termites and feldspar for the worms and ants. Was there no situation which you could also have tested the quartz and feldspar at one condition (I personally would have chosen the worms that had both mounding and subsurface mix)? I understand that you are limited by the source geology but there must have been one site/condition*

*you could have tested both? I'm sure I don't need to iterate to the authors that knowing the bleaching rates of both minerals in one condition site would be extremely useful.*

**Response:** The motivation for using either quartz and feldspar as a bioturbation tracer is elaborated in the original OSL papers (Kristensen et al., 2015; Reimann et al., 2017; Román-Sánchez et al., 2019; von Suchodoletz et al., 2023) that we used in this paper. Reimann et al. (2017) – who worked on profile SC9 within the catena of Roman-Sanchez et al. (2019) - performed a systematic single-grain quartz vs. feldspar comparisons. They concluded that single-grain feldspar measurements are not only more time- and labour-efficient in terms of data gathering but also bear the advantage of not being affected by signal sensitization during the reworking process which results in more reliable estimates of the fraction of grains in saturation, especially in settings in close proximity to plutonic and/or metamorphic bedrock. In von Suchodoletz et al. (2023) a single-grain quartz vs. feldspar comparison was performed on two samples from a Chernozem (i.e. dominated by worm bioturbation). This test resulted in agreeing quartz vs feldspar ages for both samples (their table 1). The authors concluded that feldspar single-grain measurements are ~20 times more efficient than quartz OSL single-grains and decided to focus on prior for data gathering. The results from these comparisons suggest that for theses soil settings the differential bleaching rates between quartz OSL and feldspar IRSL/pIRIR does not have significant impact on the palaeodose obtained.

*Lines 129-133. Please define soil particles or luminescence particles by some weight or dimension. At this point in the paper the reader has no idea if we are talking cm and mg or cm and g. Just give some idea of the size and weight if possible. Are they really sand sized particles?*

**Response:** These luminescence particles are not modelled as actual soil particles, in that they have a specific dimension. Rather, we consider them as objects that carry a specific age that is analogous to a luminescence age, i.e. the age accumulates when buried and resets when exposed at the surface. The model itself is mass-based and simulates the redistribution of different mass fractions in the soil. For example, with a mixing event, a certain fraction of each texture class is transported to another layer. Because luminescence is typically measured in the sand fraction, we use the ratio of transported sand as a transport probability for the luminescence particles. If needed, the transport could also be linked to other texture classes, for example when silt is targeted for luminescence dating.

We understand that this might be confusing. That is why we added the following clarification to this section: "These particles don't have a specific dimension, but should be considered as objects that carry a specific age that is analogous to a luminescence age", and we remove the notation that they act similarly as sand particles in real soils.

*Figure 2. In caption or elsewhere clarify that bioturbation from termites, worms, and ants are definitely not limited to the first meter and indeed in termites' case may be several meters. Obviously, you had to limit the depth for your model but make it clear you do understand the bioturbation processes vary in depth. Justify why you picked 1m or 1.5 m or 2 m.*

**Response:** Bioturbation is indeed not limited to 1 meter in real soils. In our simulations we use a depth of 1 meter to standardize the mixing processes, but actual active mixing zones are organism- and ecosystem-dependent. We will clarify this by removing the dimension from the Y-axis in Figure 2 and indicate the active mixing layer as zone where the mixing occurs. We will also clarify in Section 2.3.4 that we chose an active mixing depth of one meter in the model, but that this depth is different for each setting: "These parameters and active mixing depths were selected to illustrate how bioturbation affects luminescence-based depth profiles in our simulations, but should be

constrained with experimental data, or through inverse modelling, when applied to real-world settings".

*Line 175. I thought the model depth was 1 meter not 2 meters? Figure 4 shows 1.5 m depth. I have some confusion about the depth of model and measurements. Were measurements made for an entire 2 m to represent both mounding and subsurface? I missed where the mounding ends and subsurface begins. Line 184 indicates a 1 m depth.*

**Response:** The active mixing depth is defined as the upper meter of the soil. In case of mounding, all bioturbated material is sourced from this depth and transported to the surface. In case of subsurface mixing, the active mixing layer is also 1 meter, but due to the diffusive transport, material sourced in the active mixing layers is also transported to deeper layers. We also mentioned this in lines 230-231. We will clarify in Section 2.3.4 that the active mixing depth is 1 meter, but that we need a soil profile of 2 meters deep to account for the mixing due to diffusive transport.

*I found the descriptions of the model parameters in the text to be actually quite tedious and wished I had a table to refer to that I could quickly see everything described between lines 175-185. In fact such a table would have been more useful than the current Table 2 which I never looked at again. I did look at the model parameters a couple of times to remind myself and flipping back to text was….tedious. Maybe you could put it in this table the depths of the model, subsurface, and mound depths. Just we are all clear and on the same page. Maybe also set up along the lines of what you describe in lines 350-355.*

**Response:** We will add the model parameters to Table 2.

*TECHNICAL CORRECTIONS:*

*Wording is awkward here on lines 45-50 "The luminescence signal accumulates over time due to ionizing radiation emitted from radionuclides of elements within the uranium and thorium decay chains, as well as potassium-40, which are present in the soil, and due to cosmic rays". It makes it sound like cosmic rays control everything. It could be better stated as "The luminescence signal accumulates over time due to ionizing radiation emitted from radionuclides of elements which are present within the soil. The uranium and thorium decay chains, potassium-40, and cosmic rays all contribute varying amounts to luminescence growth in minerals".*

**Response:** We will adopt your suggestion, with some modifications: "The luminescence signal accumulates over time due to ionizing radiation coming from naturally occurring radionuclides in the soil (uranium and thorium decay chains and potassium-40) and from cosmic rays.

*Line 177. Two periods. Delete one.*

***Response:*** Done!

**References**

Kristensen, J. A., Thomsen, K., Murray, A., Buylaert, J.-P., Jain, M., and Breuning-Madsen, H.: Quantification of termite bioturbation in a savannah ecosystem: Application of OSL dating, Quaternary Geochronology, 30, https://doi.org/10.1016/j.quageo.2015.02.026, 2015.

Reimann, T., Román-Sánchez, A., Vanwalleghem, T., and Wallinga, J.: Getting a grip on soil reworking – Single-grain feldspar luminescence as a novel tool to quantify soil reworking rates, Quaternary Geochronology, 42, 1–14, https://doi.org/10.1016/j.quageo.2017.07.002, 2017.

Román-Sánchez, A., Reimann, T., Wallinga, J., and Vanwalleghem, T.: Bioturbation and erosion rates along the soil-hillslope conveyor belt, part 1: insights from single-grain feldspar luminescence, Earth Surface Processes and Landforms, 44, 2051–2065, https://doi.org/10.1002/esp.4628, 2019.

von Suchodoletz, H., Kühn, P., Wiedner, K., and Reimann, T.: Deciphering timing and rates of Central German Chernozem/Phaeozem formation through high resolution single-grain luminescence dating, Scientific Reports, 13, 4769, https://doi.org/10.1038/s41598-023-32005-9, 2023.

**Response referee 2**

Dear Hao Long,

Thank you for your review of our manuscript and suggested improvements. We will include them in our manuscript, following our responses below.

On behalf of all authors,

Marijn van der Meij

***General Comments:***

*This paper aims to differentiate the impacts of various mixing processes, specifically mounding and subsurface mixing, on soil profiles. The authors have compiled three luminescence-based datasets that illustrate soil mixing by different biota and compared them to numerical simulations of bioturbation using ChronoLorica, a soil-landscape evolution model. The research topic is interesting, and the methods and results demonstrate a considerable level of credibility. The paper is well-organized, with a clear logical progression and concise language. It is undoubtedly deserving of publication; however, some modifications are necessary before it can be published.*

***For lines:***

*Line 146: Based on the expression of the formula (Eq. 3-5), it seems to me that $BT_{pot}$ represents something akin to the **maximum** potential disturbance rate, while BT(z) represents the potential disturbance rate at each depth. If that's the case, it should be clarified in the text.*

**Response:** Thank you for pointing this out. We realised we had written these equations incorrectly. $BT_{pot}$ is actually the total amount of bioturbation occurring in the soil. This amount is distributed over each soil layer based on the depth function and the depth of this layer, following Eq. 6. Equations 3-5 describe these depth functions. We will rename these equations $df_{exp}(z)$, $df_{grd}(z)$ and $df_{abr}(z)$ for the different depth functions and remove the notation of $BT_{pot}$. We will explain in the text that Eqs. (3-5) are the depth functions, which are used in Eq. (6) to distribute the potential bioturbation rates across all soil layers. This also provides a more extensive explanation of Eq. (6), as is requested in a further comment.

*Line 175: What does "one-dimensional soil profiles" mean? Based on the discussion about "two to three-dimensional settings" in section 4.2, I presume that "one-dimensional" refers to considering only vertical movement of particles in the soil profile. Is that correct? Anyway, adding a sentence or two to clarify this would improve understanding.*

**Response:** The simulations are indeed performed on a single soil column or pedon, where the only dimension is depth. This allows us to focus on vertical mixing processes in the simulations, without the effects of lateral redistribution processes. We will clarify this in the text.

*Line 177: There is an extra period at the end of the sentence. Please remove one of them.*

**Response:** Done!

*Line 178: The simulations were conducted using loess-like soil texture. For the simulation itself, such simplification is, of course, reasonable. However, for the calibration as discussed in section 4.3, it*

*might be worth considering simulating it using the soil composition that closely matches the referenced profiles. This modification might also help explain a portion of the deviation observed between the simulated and expected results.*

**Response:** The bioturbation process is currently included as a uniform mixing process that does not consider different mixing rates for different texture classes. Therefore, the simulations and deviations during the calibration are not affected by the parent material composition. It is possible to include grain-size dependent mixing rates, but that is outside of the scope of this study. We will however mention this as one of the directions for future research that could improve our simulations of bioturbation and potentially the calibration process of the model as well.

*Line 194: How is the "modal age" calculated? Is it similar to the Central Age Model (CAM)? Providing more details on this would be helpful.*

**Response:** The modal age is the age corresponding to the highest peak in the age distributions, which is calculated with the Sheather and Jones method (line 191). With the highest probability, we consider this the most likely age of the sample. When the $D_e$ values are (log-)normally distributed, this age should be similar to a Central Age Model, but these ages can deviate when the type of distribution changes. The distributions in our study, both experimental and numerical, show clear peaks or modes, but are also very skewed.

The advantage of using the modal ages is that we can work with the age distributions, both from experimental data and simulated data, without the need for assumptions based on different age models and the estimation of additional statistics such as the overdispersion. This allows us to automate and speed up the calibration process without introducing uncertainties from age model selection. A disadvantage is that we don't have a robust estimate of the error of the estimated age, but for our calibration purposes this is not required. The model can be expanded with the use of age models instead when that is necessary for a specific study.

We will argue our choice of the modal age in the manuscript, provide more information on how it is calculated and add more information on the age distributions in a supplementary file, also based on the comments of reviewer 3.

*Line 205-206: The statement "with a larger contribution … decrease." compares the overall characteristics among the three profiles. However, the interquartile ranges and the bioturbated fractions vary significantly within each profile, and there is some overlap in the data range among different profiles. This general statement is therefore inaccurate and confusing. It is recommended to modify or completely remove this sentence.*

**Response:** We agree that this pattern is not clear for the interquartile range, but for the bioturbated fraction there is a clear trend with increasing subsurface mixing, especially for the subsurface. We will therefore remove the mention of interquartile range from this sentence and specify that we are talking about the subsurface.

*Line 230: It is unclear where the "increase in scatter" is reflected in the data. As the text (L195) states that the interquartile range reflects the width of the distribution, I suppose it's also a parameter capable of describing the "scatter in the age distribution". However, as the depth increases, the interquartile range curves for all functions shift to the left (towards younger ages, Figure 4B), and it's also mentioned in L232-234 that "interquartile ranges… generally decrease down the profile.". Therefore, I am confused what does the "increase in scatter" exactly mean? More explanation is need here.*

**Response:** This scatter refers to the variation in the depth profiles of the modal ages, which increases below 0.5 meters. We will clarify this in the text.

*Line 253: The description of Figure 6A is inaccurate. The statement "termites... show lower modes of ages compared to worms..." is only evident in the lower part of the graph (e.g., the range from 0.6 to 1 on the y-axis). However, in the upper part of the profile, the mode ages of termites are older or comparable to those of worms. It is mentioned in the text that the results of ants were not used for comparison due to calibration issues. Actually, the comparison between termites and worms is also questionable due to significant differences in the age used for calibration.*

**Response:** This statement indeed refers to the patterns in the subsurface, where we see lower modal ages for termites, but also for simulations with more mounding. We will clarify this in the text.

It is indeed complicated to compare absolute values of the luminescence depth profiles due to differences in age. We partly countered this by normalizing the ages, which can help us in qualitatively comparing patterns, as we did in Figure 6. For calibration purposes, (Section 4.3), these profiles are treated individually and the calibration parameters were based on the individual case studies. The ants dataset was not included in the calibration, because we expected a substantial effect of erosion processes on the luminescence depth profiles. For the termites and worms datasets, which were sampled on flat terrain, this would not play a significant role.

*Line 435: Delete one of the redundant "leading to".*

**Response:** We removed one of the repetitions.

***Regarding sections and equations:***

*Section 4.1: The discussion on the presence of other types of bioturbations is commendable. To strengthen this point, it is suggested to incorporate a bit more information about the geographical and vegetation environment of the referenced luminescence profiles, followed by further discussions on whether or not the three luminescence profiles used in the study are influenced by uprooting or other factors.*

**Response:** We will add information on the ecosystems and vegetation types of the experimental studies to Table 1, and refer to this in the Discussion on other potential mixing processes. We expect no significant contribution of uprooting, due to the natural steppe vegetation of the Chernozem from the worms site, the savannah vegetation of the termites site and lightly forested grassland in Spain from the ants site. We will add this to the Discussion in Section 4.1.

*More explanations should be made for Eq.6 and 7.*

**Response:** We added additional explanations for Eq. 6 based on an earlier comment and will explain Eq. 7 in more detail as well.

*Concerning the soil mixing tracer, I would like to suggest you refer to a recent work on mollisols from China (Zhang et al., 2024, Reconstructing Mollisol Formation Processes Through Quantified Pedoturbation, GRL; https://doi.org/10.1029/2024GL108189)*

**Response:** We will add this paper to the studies listed in the introduction.

***Regarding graphs and tables:***

*I noticed a discrepancy in the labeling of figures. The labels in the figures were marked with lowercase letters (a, b, c), while uppercase letters (A, B, C) were used in the captions and references in the text. Please make sure to check the journal requirements and consider unifying the labeling format accordingly.*

**Response:** Thank you for noticing this. We will modify it to the lowercase letters in between brackets, following the journal guidelines.

*Figure 6c, please check the x-axis title.*

**Response:** We will change the label to $f_{bio}$.

*Table 1, please check the spelling of the luminescence method used for ants "IR50IRIRSLe".*

**Response:** Thank you for spotting this. It should read post-$IR_{50}IRSL_{175}$. We corrected the spelling in Table 1.

**Response referee 3**

Dear Adrian A. Wackett,

Thank you for the extensive review of our manuscript and the valuable comments regarding soil ecology and statistics. We will consider your comments in the manuscript following our detailed responses below.

On behalf of all authors,

Marijn van der Meij

*GENERAL COMMENTS –*

*The authors have combined a preexisting soil-landscape evolution model with a new model designed to decipher soil mixing and mounding signals from luminescence depth profiles. They then use their integrated model(s) to estimate total bioturbation rates and partition the relative contributions from mounding vs. subsurface mixing for a set of empirical luminescence datasets where ants, earthworms, and termites are the dominant pedoturbation agents. The scope and depth of the manuscript is reasonable and the text itself is well written, concise, and engaging. I find the research question(s) to be both well motivated and supported by the study design/modeling framework. I believe the manuscript will make a valuable contribution to our emerging understanding of the coupling between soil mixing and soil/landscape development. I have no major issues or concerns with the framing and overall arc of the manuscript – I would mostly like to see the authors more rigorously scrutinize and discuss their model outputs/comparisons and potentially revisit some of the summary statistics/visual outputs that may be obscuring potentially important information about the (experimental or modeled) distributions of interest.*

*SPECIFIC COMMENTS –*

***Additional discussion of different species' functional types:*** *In line 94 the authors state 'anecic earthworms who both mound and mix the subsurface', but earlier in the text they introduce anecic earthworms (and earthworms more generally) as a 'type' example of subsurface mixing. These statements seem somewhat contradictory. I suggest revising the discussion of earthworms (namely within the introduction, as this is better handled in the discussion) to more clearly delineate how the different earthworm functional types (epigeic, endogeic, anecic, etc., see Marcel Bouche's original work or Bottinelli et al., 2020 for a more recent (re-)articulation) with different feeding and burrowing behaviors likely contribute variably to advective mounding vs. diffusion. In my experience different earthworm communities/functional types generate very different pedogenic outcomes. For example, purely endogeic species are seemingly the most vigorous subsurface mixers, but their continued casting at the surface may still be viewed as small-scale mounding to some degree? And this type of mounding differs significantly from that of Lumbricus terrestris (nightcrawler) middens. I would also expect similar species-specific interactions and effects for ants, given the divergent behavior between large mound building ants (e.g., Formica rufa) versus subterranean species, for example. I suppose the same goes for termites, too. Which species/functional types were dominant at the sites considered here? Considering that there are many thousands of each of these species, I think a bit more exploration and discussion of these distinct functional niches/behaviors and their documented (or hypothesized) impacts on soil mixing modes and associated luminescence signals would be helpful, both to better interpret the experimental datasets here and to more thoughtfully apply this model elsewhere.*

**Response:** We agree that more information on the different bioturbating species would benefit the manuscript. However, in this study we rely on existing experimental studies that don't always report the specific species responsible for bioturbation. For the termites study, the species is *Macrotermes natalensis*. For the worms study, only the functional type is reported (anecic earthworms), but no specific species. The same goes for the ants study, where it is reported that ants are most likely responsible for the bioturbation, but no specifics on the species are reported either. We will report the species responsible for bioturbation where available in the manuscript.

We will also revise the introduction regarding the earthworm functional types and their respective mixing behaviour. We do not think that we contradict ourselves, as the anecic earthworms both mound and subsurface mix, but we will revise that text to avoid any confusion. We will not go into too much detail regarding specific ant, termite and worm species, as we believe this would distract from the main message of the paper, which is long-term mixing patterns due to mounding and subsurface mixing.

***Changing layer/profile depths – constraining the potential importance of volumetric strain & erosion/deposition, aeolian inputs, etc:*** *Perhaps this comment pertains to the (Chrono)Lorica modeling frameowrk more generally, but I am slightly confused as to how (or whether) this model accounts for time-variant volumetric strain/collapse catalyzed by pedoturbation, or any other additions/losses of sediment (i.e., erosion, deposition, aeolian inputs or losses, etc) over the extended soil residence times? I can appreciate that fixing depths and holding BD constant is helpful and likely necessary to reduce model complexity in this context, but volumetric changes at both the individual layer and entire pedon scale are inevitable consequences of both subsurface mixing and mound construction. This seems crucial when attempting to export the coupled ChronoLorica and Mixed Signals models into real landscapes. Would such changes not be expected to significantly alter luminescence signals? What if the entire surface is aggrading or eroding? Some additional discussion of the emergent volume/depth changes resulting from bioturbation and their potential impacts on model outputs/calibration would help tie the modeling framework to real-world landscapes. Similarly, it would be nice to at least mention somewhere the hypothesized role of erosion/deposition, given its omnipresence and critical role in sculpting earth's surface. It also seems inevitable that grains are continually being introduced into and exported out of these profiles over the timescales of interest, particulary in the case of mounding (see Wilkinson et al., 2009 and Wackett et al., 2018 for links between ant mounding and sediment export), but this is largely glossed over besides a brief mention of aeolian inputs in the discussion section. I am certainly not suggesting a comprehensive modeling assessment of these different scenarios: some cursory discussion or perhaps a brief sensitivity analysis should suffice. At the very least there should be some more explicit discussion of how these factors are handled herein without forcing readers track down and read the original ChronoLorica publications and/or wade into the supplementary code.*

**Response:** The model behaviour regarding volume and mass changes due to bioturbation is briefly mentioned in lines 154-160, where we explain how changing mass in the layers leads to changes in thickness and volume and how the model numerically deals with layers that become very thick or very thin, by either splitting or merging them. The effects of volumetric strain and changes in pore structure are not considered in the model. Rather, we mainly consider the net effects of bioturbation and soil replacement on the mass composition and luminescence signals of the different layers, over millennial timescales. Including these pore size dynamics would be too complex and detailed for the spatial and temporal dimensions and the level of complexity of the ChronoLorica model. We think that for accurately considering volumetric strain, a more detailed

modelling approach, with smaller spatial and especially temporal resolution would be more suitable. We will clarify this in Section 2.3.1.

Erosion and deposition processes are also included in the ChronoLorica model (e.g. Van der Meij et al., 2023), but were not included in this study where we focus on vertical mixing patterns. However, we do comment on the effect of these processes on the luminescence-depth profiles for the ants dataset, where erosion most likely occurred (lines 288-293). For the termites and worms studies, erosion processes would not significantly impact the luminescence-depth profiles, as these study sites are located on flat terrain. Eroded termite mounds are also deposited locally and are incorporated into the active mixing zone, and can therefore be considered as one-dimensional systems as well. For our simulations, we simplified the development and erosion of surface mound into the generation of a new surface layer (lines 164-165). Román-Sanchez et al (2019, Figure 2) provide a detailed description of how luminescence-depth profiles are affected by water erosion and soil creep. We will briefly summarize this in our Discussion. Simulation of the interactions between bioturbation and erosion and their effects on luminescence signals are planned for future studies.

***Distribution statistics and accompanying visualizations:*** *These comprise the bulk of my comments, as I have a few questions/concerns about the selection of distribution statistics and would like to see some additional information included in the figures and/or text to more thoroughly explore some of the model comparisons (this same theme applies to several ensuing comments). First, why not display the median instead of the mode in the various depth figures? It appears that at least some of the soil layers display zero-inflated distributions (based on figure 3, which is the only figure that actually shows the sample point distribution – more to this point below), which could heavily bias estimation of the mode. The median should be far less sensitive to such biases. I know that the median is to some degree captured within the IQR, but it still seems like a more robust statistic. Even if there's some luminescence-specific or other reason to opt for the mode rather than the median, there should be some discussion of this as it would help readers better understand how and why you made the choices you did.*

**Response:** We will reply to all comments on statistics at the end of this part.

***Implicit/unexplored assumption of normal distributions?*** *There is quite a bit of weight placed on the IQR as a robust metric for measuring the width of the distributions. How sensitive is this metric to the type of distribution (i.e. normal, log-normal, Pareto, gamma, etc.)? The experimental datasets shown in Fig 3 seem to span not only a range of distribution widths, but also different distribution types (although this is a bit hard to discern based on the current visualizations, see comment below). Have you done any chi-squared, Kolmogorov-Smirnov (K-S) or other tests to check the data against theoretical distributions or each other? Later in section 4.3 we are left to compare the modeled vs experimental distributions based only on the summary statistics (i.e., IQR and mode). Although QQ-plots would be the most direct and informative way to compare them I can appreciate that trying to show QQ-plots between different depths would be cumbersome… but why not at the very least show the modeled distributions as faded points as was done in Fig 3? Or perhaps opt for some other graphical means that more clearly conveys information about the distribution?*

***Visualizing the distributions as a function of depth:*** *I am not a luminescence expert so perhaps the current depth plots (i.e., Figs. 3-5, 8) are standard practice and are strongly preferred by the OSL community. I personally found it somewhat difficult to keep track of the IQR vs. mode lines and interpret their respective meanings. Why not show depth-discrete box-and-whisker plots or violin plots? Either of these (particularly violin plots) would simultaneously display the IQR and median while also permitting direct visual assessment of the actual distribution (including outliers, using*

*whatever criteria made sense to set for outlier detection). The data itself could even be plotted as faded circles behind the violin/box-and-whiskers. By just plotting the IQR as the metric of choice to exemplify the distribution width there is quite a lot of additional information being cast aside…*

***In summary (for the series of comments above) --*** *The manuscript would generally benefit from more robust statistical comparisons (between different layers, modeled vs. experiment observations, etc.) rather than relying primarily on qualitative statements of difference (higher, lower, larger, etc). As mentioned above, it would be optimal to assess the complete distributions against each other rather than rely on side-by-side comparisons of summary statistics like IQR, mode, etc. Regardless of whether a summary metric or the entire distributions are being compared, such comparisons between depths, experimental vs. modeled grains, etc. should involve some sort of statistical comparison, or at the minimum more direct statements about the magnitude of difference. I'll use several sentences from the beginning of the 'Results' section (first paragraph in section 3.1, lines 204-210) as an example. Here the authors state '…the interquartile ranges increase and the bioturbated fractions decrease', or '… clear differences in the bioturbated fraction'. How much do the IQR ranges increase (numerically)? Are these differences statistically significant between adjacent layers? Or just between the top and bottom layers in the profile? Similarly, are there statistics that support the stated 'clear difference' between bioturbated fractions? What vector length between the distributions qualifies as a 'clear difference'? Something simple like an ANOVA F-test could suffice or perhaps (given the seemingly non-normal and discrete distributions) a non-parametric test like K-S or the aptly named Earth Mover's Distance could be preferable? Some Bayesian approach may work too (I am just less familiar with these myself). Any combination of these (or others not mentioned here) would help offer some numbers to more clearly articulate where the most meaningful differences and comparisons lie. I do not hope for the text to become bogged down by reporting statistical outputs like P-values, etc. and I imagine many or most of these outputs could easily be embedded within a table (residing in either the main text or SI). Also note that I referenced clips from this one paragraph to offer an example, but similar statements throughout the results and discussions sections deserve some additional statistical scrutiny.*

**Response:**

***Distribution statistics and depth plots***

Luminescence studies often use probability functions to study effects of bioturbation (e.g. Bateman et al., 2003). The shape and potential multimodality of the distributions provides information on the mixing type and intensity. The modes of these distributions represent the most common age of the sample. For the intensively mixed soils due to mounding and subsurface mixing in this study, this age corresponds to the burial age of the bioturbated particles. Any skewness or multimodality may result from occasional deep bioturbation or different inherited luminescence signals from the parent material. The median is sensitive to skewness or multimodality and will therefore not correspond to the peak in case of non-normal distributions. Therefore, we think that the mode best represents the most likely age of the sample. We will motivate this choice in Section 2.4.

The IQR is generally considered a robust parameter in the presence of outliers (Sullivan et al., 2021). That is why we used it as a measure of distribution width, or spread in the ages for each sample. By selecting one parameter, we could clearly visualize changes in depths of the luminescence distributions, which served our purpose qualitatively comparing age distributions resulting from different bioturbation processes

We think that summary statistics such as the mode, IQR and bioturbated fraction are sufficient to show the main differences in age distributions resulting from the two processes. We will argue our choice for these statistics in Section 2.4.

Visualizations using probability functions and point clouds result in very messy and unclear Figures, because there is a large number of samples, data points and simulation scenarios that are compared. The same occurs when we make boxplots or violin plots for each depth, as the information supply will be very large and create an unclear image, where we cannot compare results of the different simulation scenarios. That is why we opted for the IQR as a single parameter to represent the width of the distributions in the paper. In the new submission, we will include an Appendix with extended versions of these Figures with violin plots (based on your next comment) for readers that are interested in the specifics of the distributions.

***Statistical tests***

The purpose of this paper is to provide a comparison between luminescence-based depth functions developed by different bioturbation processes to understand general mixing patterns of different bioturbation processes. As this is a first comparison of its kind, where we compare datasets from different settings and with different ages, we intend it to be mainly qualitative and provide researchers the tools to qualitatively interpret what has happened in their soil, such as the conceptual model in Fig. 7, using the above-mentioned metrics. We also do a first attempt of calibration and quantification in Section 4.3, but that is only to showcase the potential of the approach. More thorough studies on calibration and quantification are planned for future projects.

We think that the proposed statistical tests will distract from the main qualitative message of the paper, but agree that they would be very useful for a stronger coupling between experimental and numerical datasets, for example through calibration. We will mention how these statistics could help future calibration attempts in Section 4.3.

***Time step sizes and annual mixing rates:*** *It would be nice to add some (even a brief) discussion of the different climates for the 3 case studies and their potential impact(s) on the modeled vs. measured bioturbation rates. It is certainly reasonable to calculate/discuss both the modeled and measured rates per annum, but earthworms in Germany presumably have significantly fewer days to mix soils each year than termites (or any other bioturbating agents) in Ghana, where biota are active year-round. While this is true in nature, my understanding is that the model currently treats 'time' identically between the sites, given that all the sites use the same step size (please forgive me if I'm misunderstanding something here), even if they integrate (cumulatively) over different time periods. Over thousands of years the length of this 'mixing season' may also shift dramatically at each site, but in a way that should be at least somewhat readily constrained through paleoclimate proxies. I'm thus wondering if some fractional 'mixing time' parameter reflecting the proportion of active mixing days/total elapsed time could be readily incorporated into the model? This could then flexibly carry a climate dependency signal that downstream users could easily adapt to their respective needs. My thinking is that a mixing rate of 10 kg m-2 yr-1 is much more impressive in the tundra than the tropics if the mixing agent only has two or three ice-free months to work with… I also wondered whether this could be a potential reason for the offset between the modeled and measured bioturbation rates, although this offset between the modeled vs. measured rates appears consistent irrespective of climate.*

**Response:** The annual mixing rates in this study follow from the focus on evolutionary timescales ($10^2$-$10^4$ years), for which luminescence is a suitable tracer. Luminescence cannot determine subannual dynamics of bioturbation, due to its temporal precision, but potential changes in mixing season over time could be reflected in the annual rates. When one is interested in sub-annual dynamics of bioturbation, the annual rate can be divided by the fractional mixing time, which should be determined through other methods. Otherwise, other tracers and numerical methods would be more suitable for this purpose.

***Is there a hidden discrepancy between the number of modeled vs the visualized layers?*** *In the 'Model set-up' section (line 175) the authors mention that the model simulates 200 soil layers of uniform 1 cm thickness. However, it appears (at least to me) that the ensuing figures don't depict 200 distinct layers? Are these 1cm layers somehow being aggregated into variably thick horizons across different model runs? Or are they aggregated afterwards for visualization purposes? Some explanation here would be helpful. If they are indeed aggregated in the model (to reflect the layer thicknesses in the empirical dataset, for example), then do you notice any differences in model outputs when holding everything else constant but binning the layers into different thicknesses? I ask because I have used advection-diffusion models for fallout radionuclides where there is a hidden depth dependency baked into the equation… in other words it matters whether you model fluxes in and out of layers that are 1 cm vs. 5 cm vs. 20 cm thick. Such a thickness dependence doesn't break the model per se, but it would certainly be important to explore and comment on if such a depth dependency is present.*

**Response:** Thank you for noticing this. This explanation seems to have gotten lost during the writing. We indeed aggregated the model results to reduce scatter that results from the stochastic particle transport. We aggregate them per five layers, so that their thickness resembles typical 5-cm thick OSL samples. The aggregation does not affect the depth trends of the results, just the amount of scatter. We will add this calculation step to Section 2.4.

There is actually no depth dependence in the model simulations. Simulations with thicker layers will result in similar outputs, again with lower scatter when working with thicker layers. The only depth dependence is due to the bleaching depth, which determines the depth over which particle ages are reset. As mentioned in Section 4.3, this parameter requires constraining with experimental data to improve the model.

*TECHNICAL CORRECTIONS:*

*Line 64: awkward phrasing here with 'provide a stronger role of biota…' Consider changing to 'better constrain the role of biota' or similar*

**Response:** We will rephrase this to: "better represent the role of biota in soil-landscape evolution models…".

*Line 288: missing 'a' – text should read 'with only a few luminescent…'*

**Response:** we will add the "a".

*Line 293: Note that the sites in Heimsath et al., 2002 also have significant ant and earthworm-mediated soil mixing and mounding (see Wackett et al., 2018 for discussion about the key role ants play at these sites). I would suggest modifying the wording here to be more inclusive of other mixing agents, as my impression is that root activity/tree throw is a relatively minor contributor relative to other biotic agents like earthworms, ants, wombats, etc.*

**Response:** The work of Wackett et al. (2018) actually shows the point that we want to make here, that mounded material gets removed by overland flow. We will add this reference to the discussion and mention mounding by ants and gophers at this site as well.

*Lines 304-306: Agreed! Well stated. I'm looking forward to the future work that incorporates additional mixing modes like tree throw/upheaval!*

**Response:** Thank you.

*Line 309: another missing 'a' and consider swapping 'measured' for 'used'? Text should read 'where they can be used as a tracer for soil mixing'.*

**Response:** We adopted both suggestions.

*Line 435: delete the extra 'leading to'*

**Response:** Done!

*Line 440: either say '…due to the overestimation of…', or alternatively delete 'the' so it instead reads '…due to overestimating the…'*

**Response:** We removed the "the".

**References**

Bateman, M. D., Frederick, C. D., Jaiswal, M. K., and Singhvi, A. K.: Investigations into the potential effects of pedoturbation on luminescence dating, Quaternary Science Reviews, 22, 1169–1176, https://doi.org/10.1016/S0277-3791(03)00019-2, 2003.

Román-Sánchez, A., Reimann, T., Wallinga, J., and Vanwalleghem, T.: Bioturbation and erosion rates along the soil-hillslope conveyor belt, part 1: insights from single-grain feldspar luminescence, Earth Surface Processes and Landforms, 44, 2051–2065, https://doi.org/10.1002/esp.4628, 2019.

Sullivan, J. H., Warkentin, M., and Wallace, L.: So many ways for assessing outliers: What really works and does it matter?, Journal of Business Research, 132, 530–543, https://doi.org/10.1016/j.jbusres.2021.03.066, 2021.

Van der Meij, W. M., Temme, A. J. A. M., Binnie, S. A., and Reimann, T.: ChronoLorica: introduction of a soil–landscape evolution model combined with geochronometers, Geochronology, 5, 241–261, https://doi.org/10.5194/gchron-5-241-2023, 2023.

Wackett, A. A., Yoo, K., Amundson, R., Heimsath, A. M., and Jelinski, N. A.: Climate controls on coupled processes of chemical weathering, bioturbation, and sediment transport across hillslopes, Earth Surface Processes and Landforms, 43, 1575–1590, https://doi.org/10.1002/esp.4337, 2018.

---

## Author Response (AR2)

Dear editors,

Thanks for accepting our paper in SOIL. We have made the edit suggested by reviewer 2 in lines 335-336. Next to that, we also added some missing DOIs to the references.

The last thing we need to do is to upload the accompanying model code to ZENODO and reference it in the manuscript. For this, I would like to know the DOI of the paper, so that I can refer to it in the model code. Could you let me know when the DOI will be available?

With best regards,

Marijn van der Meij